# ppGpp accumulation reduces the expression of the global nitrogen homeostasis-modulating NtcA regulon by affecting 2-oxoglutarate levels

Ryota Hidese[1,2], Ryudo Ohbayashi[3,7], Yuichi Kato [2], Mami Matsuda[1], Kan Tanaka[3], Sousuke Imamura[3,4], Hiroki Ashida[5], Akihiko Kondo [1,2,6] & Tomohisa Hasunuma [1,2,6 ✉]

The cyanobacterium *Synechococcus elongatus* PCC 7942 accumulates alarmone guanosine tetraphosphate (ppGpp) under stress conditions, such as darkness. A previous study observed that artificial ppGpp accumulation under photosynthetic conditions led to the downregulation of genes involved in the nitrogen assimilation system, which is activated by the global nitrogen regulator NtcA, suggesting that ppGpp regulates NtcA activity. However, the details of this mechanism have not been elucidated. Here, we investigate the metabolic responses associated with ppGpp accumulation by heterologous expression of the ppGpp synthetase RelQ. The pool size of 2-oxoglutarate (2-OG), which activates NtcA, is significantly decreased upon ppGpp accumulation. De novo $^{13}$C-labeled $CO_2$ assimilation into the Calvin-Benson-Bassham cycle and glycolytic intermediates continues irrespective of ppGpp accumulation, whereas the labeling of 2-OG is significantly decreased under ppGpp accumulation. The low 2-OG levels in the RelQ overexpression cells could be because of the inhibition of metabolic enzymes, including aconitase, which are responsible for 2-OG biosynthesis. We propose a metabolic rearrangement by ppGpp accumulation, which negatively regulates 2-OG levels to maintain carbon and nitrogen balance.

[1] Graduate School of Science, Innovation and Technology, Kobe University, 1-1 Rokkodai, Nada, Kobe 657-8501, Japan. [2] Engineering Biology Research Center, Kobe University, 1-1 Rokkodai, Nada, Kobe 657-8501, Japan. [3] Laboratory for Chemistry and Life Science Institute of Innovative Research, Tokyo Institute of Technology, Midori-ku, Yokohama, Japan. [4] NTT Space Environment and Enegy Laboratories, Nippon Telegraph and Telephone Corporation, 3-9-11 Midori-cho, Musashino-shi, Tokyo 180-8585, Japan. [5] Graduate School of Human Development and Environment, Kobe University, 3-11 Tsurukabuto, Nada-Ku, Kobe 657-8501, Japan. [6] Research Center for Sustainable Resource Science, RIKEN, 1-7-22 Suehiro, Tsurumi, Yokohama, Kanagawa 230-0045, Japan. [7] Present address: Department of Biological Sciences, Faculty of Science, Shizuoka University, 836 Ohya, Suruga-ku, Shizuoka 422-8529, Japan. ✉ email: hasunuma@port.kobe-u.ac.jp

Environmental stresses incur a broad range of physiological changes in the majority of bacterial species. This stringent response, which is mediated by the alarmone guanosine penta- or tetraphosphate known as (p)ppGpp, is important for survival during environmental stress[1–4]. (p)ppGpp is biosynthesized by the (p)ppGpp synthetase SpoT/RelA (RSH), which catalyzes the transfer of the adenosine 5'-triphosphate (ATP) pyrophosphate moiety to the 3'-OH position of guanosine 5'-triphosphate (GTP) or guanosine 5'-diphosphate (GDP) ribose[5–7]. Moreover, the intracellular accumulation of (p)ppGpp leads to global transcriptional reprogramming, which controls the repression of genes involved in macromolecular biosynthesis, cell division, and motility; and the induction of genes responsible for amino acid biosynthesis, nutrient acquisition, and stress response.

Cyanobacteria, including the oxygenic photosynthetic cyanobacterium *Synechococcus elongatus* PCC 7942, transiently accumulate (p)ppGpp during light–to–dark transition or nitrogen starvation[8–11]. In the filamentous cyanobacterium *Anabaena* spp., ppGpp accumulates during the early stages of heterocyst differentiation under nitrogen starvation conditions[12,13]. Dark conditions limit the availability of external energy sources for cyanobacteria, leading to metabolic rearrangement, i.e., changes in carbon and nitrogen fixation pathways[14]. Higher plants also use (p)ppGpp signal transduction to survive under stress conditions, such as drought, UV irradiation, and nitrogen starvation, by accumulating (p)ppGpp in their chloroplasts[15–17]; this suggests the evolutionary relevance of response signaling in cyanobacteria. *S. elongatus* PCC 7942, which regulates intracellular (p)ppGpp levels via a single RSH protein, Rel, and is responsible for its synthesis and hydrolysis.

A previous study reported that (p)ppGpp signaling in *S. elongatus* PCC 7942 contributes to dark adaptation via alterations in cellular physiology, including the control of photosynthetic pigments and polyphosphate levels, DNA content, and translation rate[18]. (p)ppGpp plays a crucial role in maintaining viability during darkness[18]. The (p)ppGpp-deficient strain exhibited increased transcription and translation rates during photosynthetic growth and impaired viability during the dark period[19]. These results indicated that (p)ppGpp is important for adaptation under both light and dark conditions. On the other hand, the overexpression of (p)ppGpp synthetase from *Bacillus subtilis* YjbM/SAS1/RelQ[20] causes apparent phenotypic responses, such as growth arrest, loss of photosynthetic activity and viability, increased cell size, and accumulation of large polyphosphate granules[18,21]. In addition, (p)ppGpp accumulation upregulates ribosomal hibernation-promoting factor, which causes ribosomes to dimerize to decrease translation rates of protein synthesis machinery[18]. Moreover, previous comprehensive transcriptome studies, using RNA-sequencing (RNA-seq), revealed that artificial (p)ppGpp under light conditions upregulates *gifA* and *gifB*, which encode glutamine synthetase inactivating factors, but downregulates various other genes, including the nitrite reductase gene *nirA*, which is responsible for nitrogen assimilation[18]. RelQ overexpression does not reflect the induction and quantitative changes in (p)ppGpp in response to environmental changes, and the observed cellular responses would actually show over-responses. However, the cellular responses by RelQ over-expression provide crucial insight into the mechanisms of action of (p)ppGpp. *gifA* and *gifB* genes are downregulated, whereas *nirA* is upregulated by the transcription factor NtcA[22,23]. Cyanobacteria rapidly activate the nitrogen-assimilation pathway under nitrogen starvation conditions by sensing increasing levels of the major nitrogen limitation signal molecule 2-oxoglutarate (2-OG)[24–27]. A decrease in 2-OG levels negatively regulates the DNA-binding activity of the global nitrogen regulator NtcA, resulting in the inactivation of nitrogen metabolism[28–31]. These findings suggest an important link between NtcA activity and (p)ppGpp accumulation. However, mechanistic insights into the changes in transcript abundances of these specific genes have not been examined.

In the present study, we heterologously expressed the (p)ppGpp synthetase RelQ in *S. elongatus* PCC 7942 and investigated the metabolic responses to ppGpp accumulation, using our dynamic metabolome analysis approach. The present data clearly demonstrate that ppGpp accumulation results in a decrease in intracellular 2-OG levels by repressing the de novo synthesis of 2-OG. We propose a novel model for ppGpp signaling that negatively controls the increase in intracellular 2-OG levels.

## Results

**ppGpp is involved in the regulation of genes under the control of NtcA.** To visualize the responses associated with ppGpp accumulation, *B. subtilis relQ* (previously known as *yjbM*), which encodes an RSH protein that irreversibly catalyzes the synthesis of (p)ppGpp from GTP or GDP[20], was introduced into the *S. elongatus* PCC 7942 genome under the control of an isopropyl β-D-1-thiogalactopyranoside (IPTG)-inducible promoter (Supplementary Fig. 1). The resultant RelQ-expressing cells (RelQ cells) were cultivated for 3 d at 30 °C in the presence of 1% (v/v) $CO_2$ and 17 mM sodium nitrate under photoautotrophic conditions, before IPTG addition. In the RelQ cells treated with IPTG (RelQ-ox cells), the intracellular concentration of ppGpp peaked at 24 h after IPTG addition and was 0.11 µmol/g-dry cell weight (DCW); thereafter, the concentration gradually decreased to 0.08 µmol/g-DCW (Fig. 1a). We did not observe increase in ppGpp levels in the RelQ cells without IPTG treatment. The optical density of RelQ-ox cells at 750 nm ($OD_{750}$) increased until 168 h of total cultivation (Supplementary Fig. 2a), but the cell number did not increase after IPTG addition. Moreover, the cell size increased after RelQ induction (Supplementary Fig. 2b), suggesting that ppGpp accumulation did not inhibit cellular metabolism. Next, we investigated the effect of ppGpp accumulation on the uptake of nitrogen because RelQ-ox cells exhibited the symptoms of chlorosis, a typical nitrogen-starved phenotype. The addition of IPTG repressed sodium nitrate uptake (Fig. 1b). Phenotypic responses, including increased cell size and chlorosis due to ppGpp accumulation, were similar to those observed in the RelQ-ox cells supplied with air under constant illumination (50 µmol photons $m^{-2} s^{-1}$) in a previous study[21]. Glycogen accumulation, another nitrogen-starved phenotype, was also observed in the RelQ-ox cells, with the glycogen content reaching 37% (total glycogen/dry cell weight) at 120 h (Supplementary Fig. 2c, d). The phenotypes observed in the RelQ cells cultured in sodium nitrate-supplemented medium were also observed in the RelQ cells cultured in the presence of ammonium chloride (Supplementary Fig. 3a–c), suggesting that ppGpp accumulation leads to the inactivation of these nitrogen transport systems and the nitrogen assimilation pathway.

We performed RNA-seq of RelQ-ox and RelQ cells to determine the cause of decrease in sodium nitrate uptake. We cultured the RelQ cells in the presence of 17 mM sodium nitrate at 30 °C under 1% (v/v) $CO_2$ and photoautotrophic conditions; this was followed by the addition of IPTG after 72 h, and the cells were harvested at 6 and 24 h after IPTG addition. We identified differentially expressed genes (DEGs) between RelQ-ox and RelQ cells using RNA-seq analysis and filtered them as per the following threshold values: *p*-value ≤ 0.05 and $Log_2$ |fold change (FC)| ($Log_2FC$) ≥1. In Fig. 1c, a scatter plot of $log_2$ fold-change versus the average expression signal of transcripts per million kilobases (TPM) is presented. Expression data for all genes with ordered locus names are presented in Supplementary

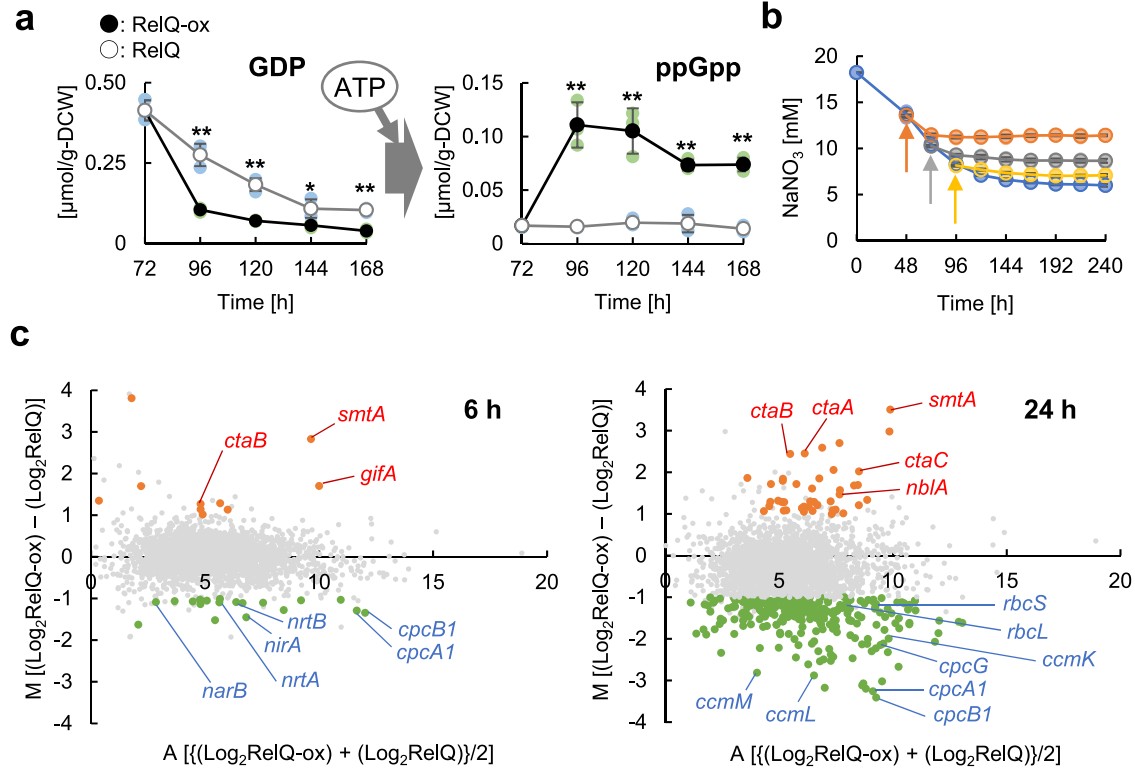

**Fig. 1 Nitrogen uptake and transcriptional dynamics upon ppGpp accumulation.** RelQ catalyzes the ATP-dependent conversion of GDP to ppGpp. **a** RelQ cells with (●, RelQ-ox) or without (○, RelQ) 1 mM IPTG addition after 72 h of initial culturing in the presence of 17 mM sodium nitrate and 1% (v/v) $CO_2$ at different time-points. GDP and ppGpp levels were measured using a capillary electrophoresis time-of-flight mass spectrometry (CE-TOFMS) system. Statistical significance was determined using Student's $t$ test (**$p < 0.01$, *$p < 0.05$). **b** Nitrate uptake response to ppGpp accumulation at different time-points. Each colored arrow indicates the timing of 1 mM IPTG addition as the final concentration (without IPTG, blue line; 2 d later, orange line; 3 d later, gray line; 4 d later, yellow line). **c** Changes in mRNA expression in RelQ cells. RNA-seq was performed for cells treated under the following conditions: RelQ cells after 6 h or 24 h with or without 1 mM IPTG addition. All panels present MA-plots (fold change vs. average) based on transcripts per million (TPM) normalization. Orange or green dots, with the gene name indicated, represent genes detected as differentially expressed with $p \leq 0.05$ and $Log_2FC \geq 1$, respectively. Genes that were not differentially expressed are colored in gray. Data are means of two biologically independent experiments.

Data 3. We identified 10 upregulated genes, including the glutamine synthetase inactivating factor IF7 gene *gifA*, and 20 downregulated genes, including the phycocyanin subunit genes *cpcA1B1*, nitrite transporter genes *nrtAB*, nitrate reductase gene *narB*, and nitrite reductase gene *nirA*, in the RelQ-ox cells 6 h after IPTG addition compared to that in the RelQ cells. Moreover, 268 genes were downregulated in the RelQ-ox cells 24 h after IPTG addition, including genes related to photosystems, light-harvesting antenna (*cpcA1B1G*), chlorophyll synthesis, $CO_2$ assimilation (ribulose 1,5-bisphosphate carboxylase/oxygenase genes *rbcL*/*rbcS* and carboxysome shell protein genes *ccmMLK*), and ATP synthesis; 43 genes, including the phycobilisome degradation protein gene *nblA* and, metallothionein gene *smtA* and *ctaABC* genes for cytochrome c oxidases and maturation, were upregulated in the RelQ-ox cells 24 h after IPTG addition. The changes in transcript abundance were also found in the previously reported transcriptional profiles of ppGpp-accumulated cells under photoautotrophic conditions with atmospheric $CO_2$, but the upregulation of genes for $CO_2$ assimilation was not observed[18]. The transcript abundance of *relQ*, which encodes intrinsic ppGpp synthetase/hydrolase (*rel*: $Log_2FC = 0.14$ at 6 h and $Log_2FC = -0.12$ at 24 h) was not significantly changed. The ppGpp-dependent transcript dynamics of several genes, including *nblA*, *nirA*, *gifA*, *ccmM*, and *ccmL*, were analyzed using quantitative real-time PCR (qRT-PCR) (Supplementary Fig. 4a). The expression of *nblA* and *gifA* increased in the RelQ-ox cells. In contrast, the expression of *ccmL*

and *ccmM*, encoding carboxysome shell proteins involved in $CO_2$ assimilation, and *nirA* decreased in the RelQ-ox cells. The changes in the expression levels of these genes as per qRT-PCR analysis were similar to those observed by RNA-seq analysis. The transcription of *gifA* and *nirA* was mainly regulated by NtcA[22,23] (Supplementary Fig. 4b), whereas *ntcA* expression did not significantly change upon IPTG addition (*ntcA*: $Log_2FC = 0.02$ at 6 h and $Log_2FC = -0.30$ at 24 h) (Supplementary Data 3), suggesting that ppGpp inhibited NtcA function.

**Effect of ppGpp accumulation on primary metabolites**. The changes in the transcript abundance of the genes under the control of the NtcA regulatory system in the RelQ-ox cells were considered the direct or indirect consequence of ppGpp accumulation because the NtcA-dependent transcript abundance is downregulated under low carbon to nitrogen ratio (low 2-OG levels) and low energy state (high adenosine diphosphate (ADP) to ATP ratio)[23,25,29]. To elucidate the metabolic rearrangement caused by ppGpp accumulation, we examined the time-course changes in intracellular primary metabolite levels in the RelQ-ox cells. We determined the pool sizes of nucleotides (Supplementary Fig. 5a), free amino acids (Supplementary Fig. 5b, c), hexose and triose phosphates, tricarboxylic acid (TCA) cycle intermediates, and Calvin-Benson-Bassham (CBB) cycle intermediates (Fig. 2). The amount of metabolites was determined by calculating their respective peak intensity areas using capillary electrophoresis time-of-flight mass spectrometry (CE-TOFMS)[32].

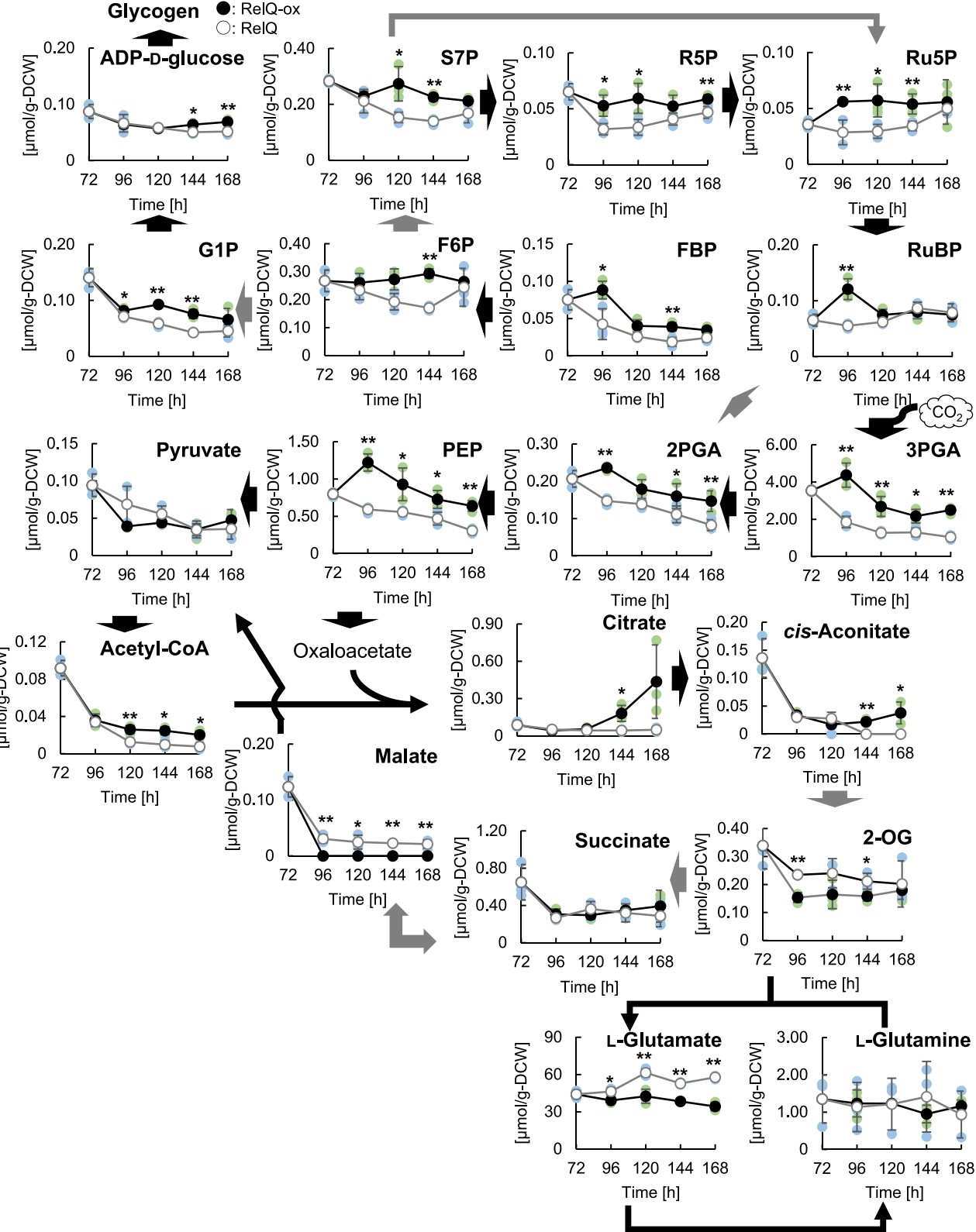

**Fig. 2 Metabolic profiling of RelQ cells.** Time-course changes in intracellular metabolite pool sizes of RelQ cells with (●, RelQ-ox) or without (○, RelQ) IPTG addition after 72 h of initial culture in the presence of sodium nitrate. Statistical significance between each sampling point of RelQ-ox and RelQ cells was determined using Student's *t* test (**$p < 0.01$, *$p < 0.05$). Black and gray arrows represent direct and indirect reaction, respectively. Abbreviations: G1P, glucose 1-phosphate; S7P, sedoheptulose 7-phosphate; R5P, Ribose 5-phosphate; FBP, fructose 1,6-bisphosphate; Ru5P, ribulose 5-phosphate; F6P, fructose 6-phosphate; RuBP, ribulose 1,5-bisphosphate; PEP, phosphoenolpyruvate; 2PGA, 2-phosphoglycerate; 3PGA, 3-phosphoglycerate; 2-OG, 2-oxoglutarate.

RelQ-ox cells exhibited larger pool sizes of various metabolites, including hexose phosphates (fructose 1,6-bisphosphate, fructose 6-phosphate, and glucose 1-phosphate), triose phosphates (2-phosphoglycerate, phosphoenolpyruvate, and 3-phosphoglycerate), the CBB cycle intermediates (sedoheptulose 7-phosphate, ribose 5-phosphate, ribulose 5-phosphate, and ribulose 1,5-bisphosphate), and acetyl-CoA than the RelQ cells during 96–168 h of culture (Fig. 2), suggesting reduction in carbon sources in the primary metabolic pathway. Citrate and cis-aconitate gradually accumulated in the RelQ-ox cells during 120–168 h of culture. In contrast, smaller pool size of the TCA metabolite 2-OG was observed in the RelQ-ox cells; the size of the 2-OG pool in RelQ-ox cells was 65% smaller than that in the RelQ cells at 96 h. Intracellular L-glutamate levels significantly decreased in the RelQ-ox cells during cultivation (Fig. 2). The small pool size of L-glutamate could be attributed to the pool size of 2-OG as the precursor metabolite in RelQ-ox cells, as the coupled glutamine synthetase (GS)/glutamine-oxoglutarate amidotransferase (GOGAT) cycle represents the dominant metabolic route for 2-OG metabolism among cyanobacteria[33,34]. The pool size of ATP, another RelQ substrate, was significantly larger in the RelQ-ox cells than in the control cells after 120 h (Supplementary Fig. 5a). The cellular energy charge in the RelQ-ox cells, calculated as the ratio of ATP and ADP + ATP, after 168 h of cultivation was two-fold higher than that of the RelQ cells. High cellular energy charge was also observed in the RelQ-ox cells cultured with ammonium chloride, and the 2-OG level was approximately 17% lower than that in the RelQ cells at 96 h (Supplementary Fig. 3d). These results suggest that ppGpp accumulation lowers intracellular 2-OG levels, leading to the suppression of NtcA function.

**ppGpp negatively affects turnover of 2-OG, but not that of glycolytic cycle intermediates**. As shown in Fig. 2, the changes in the intracellular levels of the CBB, glycolytic, and TCA cycle intermediates, including 2-OG, can be attributed to both de novo synthesis from $CO_2$ and catabolism. We investigated the de novo synthesis of the glycolytic and TCA cycle intermediates in the RelQ cells using in vivo $^{13}C$-labeling (Fig. 3). To label the newly synthesized metabolites from $CO_2$, the RelQ cells were supplied with $NaH^{13}CO_3$ after 24 h of culture, followed by IPTG addition after 72 h. The $^{13}C$ fraction, defined as the ratio of $^{13}C$ to total carbon in each metabolite, was calculated based on the mass isotopomer distributions[35].

The $^{13}C$ fractions of the hexose phosphates, G1P (63.4%), G6P (61.9%), and F6P (62.5%), and the triose phosphates, 3PGA (42.5%), 2PGA (65.9%), and PEP (68.3%), were the highest within 2 h of labeling of the RelQ-ox cells (Fig. 3). These $^{13}C$-labeling patterns were similar to those observed in the RelQ cells, indicating de novo $CO_2$ assimilation into the CBB cycle and glycolysis. The $^{13}C$-fraction of 2-OG significantly decreased in the RelQ-ox cells (1.6% at 6 h and 4.9% at 24 h) compared to that in the RelQ cells (12.5% at 6 h and 15.2% at 24 h), but that of citrate was almost similar between the RelQ and RelQ-ox cells. The initial $^{13}C$ fraction of succinate also decreased in the RelQ-ox cells; thereafter, the $^{13}C$ fraction was similar to that of the RelQ cells at 120 h. The signals of the other TCA cycle intermediates (fumarate, malate, and cis-aconitate) were below the detection limit. These results clearly demonstrated that ppGpp inhibits the de novo synthesis of 2-OG.

To investigate the effect of ppGpp on 2-OG biosynthesis, we evaluated the activities of the enzymes involved in the oxidative branch of the TCA cycle, including aconitase and isocitrate dehydrogenase, in both RelQ and RelQ-ox cells. The citrate synthase activity of the RelQ-ox cells was comparable to that of the RelQ cells, which is consistent with the $^{13}C$-fraction patterns of citrate. The isocitrate dehydrogenase activity was also similar in both types of cells (Fig. 4a). However, aconitase activity was lower in the RelQ-ox cells than in the RelQ cells. In contrast, the transcript abundance of the aconitase gene (Synpcc7942_0903) was not significantly altered by ppGpp accumulation ($Log_2FC = -0.11$). Next, we examined the inhibitory effect of ppGpp on the activity of the aconitase homolog Synpcc7942_0903, which shares 77% identity with the aconitase Slr0665 of Synechocystis sp. PCC 6803[36]. Recombinant Synpcc7942_0903 was expressed in Escherichia coli cells and purified under anaerobic conditions. The aconitase activity of Synpcc7942_0903 decreased with the addition of ppGpp, but the effect was not significant. The maximum inhibitory activity of ppGpp was approximately 63% of that observed in the absence of ppGpp (specific activity = 215 nmol/min/mg) (Fig. 4b).

**ppGpp represses de novo biosynthesis of several amino acids**. 2-OG is synthesized not only by de novo synthesis via the TCA cycle but also by amino acid transamination during amino acid biosynthesis[37]. We investigated the de novo synthesis of amino acids in the RelQ cells using in vivo $^{13}C$-labeling (Fig. 3). The $^{13}C$ fractions of L-serine, L-tyrosine, and L-alanine in the RelQ-ox cells at 24 h after labeling were 39.2%, 33.2%, and 5.5%, respectively (Fig. 5). These values were higher than those of the RelQ cells. The $^{13}C$ fractions of glycine, L-aspartate, L-histidine, and L-glutamine in the RelQ-ox cells were comparable to those of the RelQ cells at 24 h after labeling. For other amino acids (L-tryptophan, L-phenylalanine, L-leucine, L-valine, L-proline, L-arginine, L-lysine, L-isoleucine, L-threonine, L-asparagine, and L-glutamate), the $^{13}C$ fractions significantly decreased by RelQ expression during 24 h of cultivation. The low $^{13}C$ fraction of L-glutamate could be attributed to impairment of the de novo synthesis of 2-OG, as the precursor molecule, from $CO_2$ in the RelQ-ox cells. L-cysteine and L-methionine levels were below the detection limit in the present experiment. These results suggest that ppGpp accumulation inhibits the de novo synthesis of many amino acids; however, it does not inhibit the biosynthesis of all amino acids because of higher $^{13}C$ fractions of L-serine, L-tyrosine, and L-alanine observed in the RelQ-ox cells. Because the transcript abundance of the genes involved in putative amino acid biosynthesis did not significantly change by ppGpp accumulation ($Log_2FC) < 1$) (Supplementary Data 3), amino acid biosynthesis dependent on ppGpp could have been mainly regulated at the posttranslational level. Next, we examined the pool size of free amino acids in the RelQ-ox cells. ppGpp induction resulted in the accumulation of various amino acids within 24 h, except for L-glutamate (Supplementary Fig. 5b). Consistently, the concentrations of approximately all extracellular amino acids in the RelQ-ox cells were significantly higher than those in the RelQ cells (Supplementary Fig. 5c). These results suggest that amino acids are mainly produced through proteolysis owing to the expression of the small proteolysis adapter NblA[28].

**ppGpp controls L-glutamate synthesis**. We hypothesized that the inhibition of de novo 2-OG synthesis by ppGpp in the RelQ-ox cells reflects an artificial cellular response that may not occur in the wild-type cells in terms of ppGpp levels. ppGpp accumulates in S. elongatus PCC 7942 cells after being transferred to the dark[18,19]. However, the effects of ppGpp on metabolism under dark conditions are unknown. We examined the effect of ppGpp accumulation on 2-OG synthesis in the dark. The wild-type and ΔRel (rel knocked-out) strains, cultured under photosynthetic condition, were transferred to the dark and cultured for 120 min. We investigated the levels of intracellular metabolites 2-OG, L-glutamate, and ppGpp in the cells at each time-point under dark conditions,

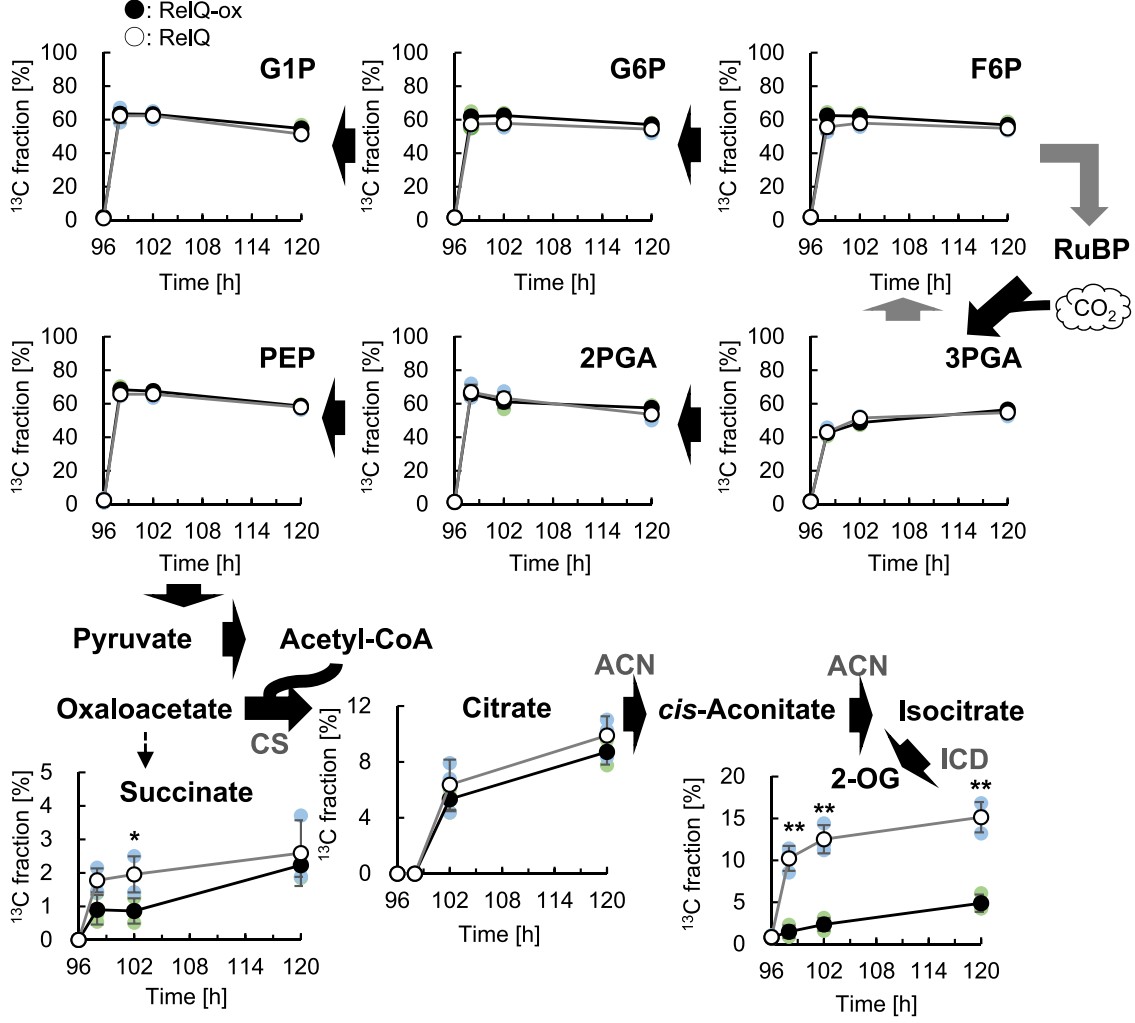

**Fig. 3 $^{13}$C-labeling pattern of the glycolytic and TCA cycle metabolites in RelQ cells.** Time-course changes in the $^{13}$C fractions of metabolites, which was initiated by the addition of 25 mM sodium bicarbonate-$^{13}$C to RelQ cell cultures after 24 h of culture in the presence of sodium nitrate, followed by the addition of 1 mM IPTG (●, RelQ-ox) or without IPTG (○, RelQ) after 72 h. Black and gray arrows represent direct and indirect reaction, respectively. All data are presented as mean ± SD (n = 3 independent biological experiments). Statistical significance between each sampling point of RelQ-ox and RelQ cells was determined using Student's *t* test (**$p < 0.01$). Abbreviations: CS citrate synthase, ACN aconitase, ICD isocitrate dehydrogenase.

and before and after the transfer of the cells to the dark, using CE-TOFMS analysis (Fig. 6). ppGpp production was not observed in the ΔRel cells, whereas the ppGpp levels in the wild-type cells increased to 0.13 μmol/g-DCW at 90 min after transition to the dark, which is comparable to the highest pool size of ppGpp in the RelQ-ox cells (Fig. 1). The 2-OG profile of the ΔRel cells was comparable to that of the wild-type cells. This could be explained by the retardation of 2-OG metabolism and limitation of $CO_2$ fixation and/or accumulation of glycolytic cycle intermediates under dark conditions[38], resulting in decreased de novo synthesis of 2-OG. Moreover, the L-glutamate levels in the ΔRel cells were significantly higher than those in the wild-type cells, irrespective of light-to-dark transition, which is consistent with the lower levels of L-glutamate in the RelQ-ox cells compared with those of the RelQ cells (Fig. 2). These results suggest that ppGpp is involved in controlling intracellular L-glutamate levels by regulating 2-OG biosynthesis, especially under photosynthetic conditions.

## Discussion

In the present study, RelQ overexpression exerted prolonged effects on the metabolism dynamics of *S. elongatus* PCC 7942

cells during ppGpp accumulation. The data are summarized in Fig. 7. Previous study reported that (p)ppGpp controls many fundamental cellular processes, including transcription, translation, DNA replication, cell growth and division, polyphosphate granule formation, and photosynthesis in *S. elongatus* PCC 7942[18]. We here revealed that artificial ppGpp accumulation dramatically alters pool sizes of metabolic intermediates, affecting some aspects of those cellular processes, including transcription of the NtcA regulon. We elucidated that ppGpp accumulation lowered the intracellular 2-OG levels by blocking the biosynthesis of 2-OG under photosynthetic conditions. Moreover, higher levels of L-glutamate were observed in the ppGpp-deficient mutant ΔRel strain than in the wild-type cells. These results suggest that ppGpp is involved in metabolic rearrangement through the repression of a key nitrogen signal—2-OG. We note that the observed phenotypes by ppGpp accumulation under photosynthetic condition must be different from ppGpp signaling under light-to-dark transition, which is a major inducer of ppGpp accumulation under physiological conditions. In the present RelQ overexpression system, it is difficult to mimic the induction and quantitative changes in ppGpp as natural responses to environmental changes. However, it is possible that if 2-OG increases too

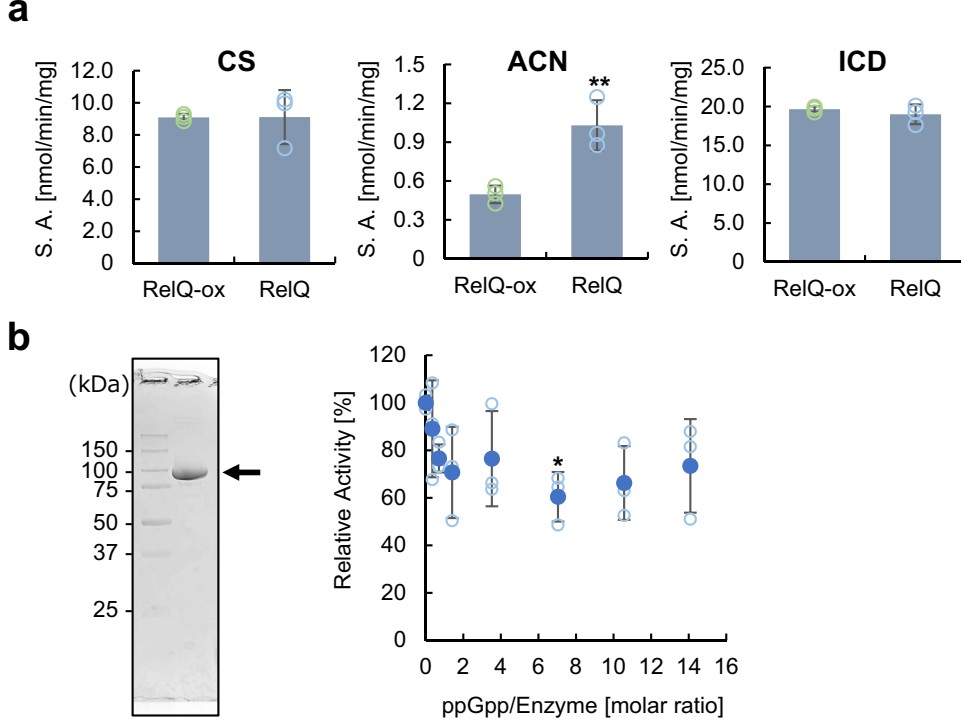

**Fig. 4 Activities of enzymes involved in the oxidative branch of the TCA cycle in the RelQ cells. a** Specific activities (SA) of citrate synthase (CS), aconitase (ACN), and isocitrate dehydrogenase (ICD) in crude extracts of RelQ cells cultured for 24 h with or without IPTG; IPTG was added after 72 h of initial culture. Statistical significance between each sampling point of RelQ-ox and RelQ cells was determined using Student's $t$ test (**$p < 0.01$). All data are presented as mean ± SD ($n = 3$ independent biological experiments). **b** (Left) SDS-PAGE of purified recombinant Synpcc7942_0903 protein, indicated by an arrow. (Right) Dose-dependent effect of ppGpp on the aconitase activity of the recombinant Synpcc7942_0903 protein. Aconitase activity of the purified recombinant proteins (0.71 μM) evaluated in the presence of various concentrations of ppGpp (0.25, 0.5, 1.0, 2.5, 5, 7.5, and 10 μM) under anaerobic condition. Activity without ppGpp was set as 100%. Statistical significance among the relative activities without ppGpp and with each concentration of ppGpp was determined using the Dunnett test (*$p < 0.05$).

much above a certain threshold, ppGpp may show a feedback effect. Indeed, this threshold is rarely exceeded in the dark, but ppGpp may play a role in preventing 2-OG levels from being exceeded during nitrogen deprivation or another starvation stress. Our data suggest that ppGpp is involved in the maintenance of 2-OG metabolism in a steady-state during photosynthetic growth. A previous study proposed that ppGpp functions in metabolic rebalancing, such as maintaining the levels of cyanophycin, a nitrogen/carbon reserve polymer, in filamentous cyanobacterium *Anabaena* sp. PCC 7120 under nitrogen starvation[13]. The inhibition of enzymatic activity by ppGpp would immediately represses metabolism to suppress the consumption of excess energy and maintain metabolic balance suitable for photosynthetic growth. Moreover, (p)ppGpp regulates gene expression and enzymatic activity by direct binding of target proteins in *E. coli*[39,40]. More than 50 putative ppGpp targets, including enzymes involved in glycolysis, the TCA cycle, and purine nucleotides, were identified using capture-compound mass spectrometry.

2-OG is synthesized through the TCA cycle, amino acid biosynthesis, and glutamate dehydrogenase[37]. The [13]C-labeling rate of 2-OG in the RelQ cells was 10-folds higher than that in the RelQ-ox cells; however, the relative activity of aconitase, which is responsible for 2-OG biosynthesis, decreased to ~60% in the presence of ppGpp. These results suggest that the total [13]C fraction of 2-OG could be contributed by amino acid biosynthesis and the action of glutamate dehydrogenase, in addition to the TCA cycle. The low [13]C fractions of these amino acids could contribute to blocking 2-OG biosynthesis.

Although RelQ-ox cells exhibit a nitrogen starvation-like phenotype, characterized by chlorosis and glycogen accumulation[41], the mechanisms underlying the ppGpp accumulation phenotype could be different from those under nitrogen-limited conditions. Cyanobacteria sense key metabolic signals 2-OG and ATP to balance the ratio of intracellular carbon to nitrogen metabolite levels and cellular energy charge, respectively. However, cyanobacteria express *nblA* through NtcA regulation to degrade phycobilisomes under nitrogen-limited conditions[24,28] and divert newly fixed carbon toward glycogen synthesis[42–44]. The omics data from our study revealed that ppGpp accumulation upregulated *nblA* without an increase in intracellular 2-OG levels, suggesting NtcA-independent activation in the RelQ-ox cells. Furthermore, the transcript abundance of transcription factors NblR, NblC, and Ald, which are responsible for the induction of *nblA* under nutrient starvation[25], did not change upon IPTG addition (Supplementary Data 3). In contrast, *smtA*, encoding metallothionein, and *ctaBAC*, encoding the components of cytochrome c oxidase, were significantly upregulated upon ppGpp accumulation. These data suggest that ppGpp causes an imbalance in the redox status via the downregulation of the nitrogen assimilation system. The dysfunction of nitrogen assimilation due to ppGpp accumulation causes NblA expression, which could be due to over-reduction of the photosystem[45]. Meanwhile, an oversupply of ATP could lead to glycogen accumulation, as cell growth is hampered by ppGpp accumulation. The extra energy charge contributes to the promotion of ATP-dependent cellular activities, such as Clp proteases associated with NblA[46] and the energy-dependent excretion of amino acids[47].

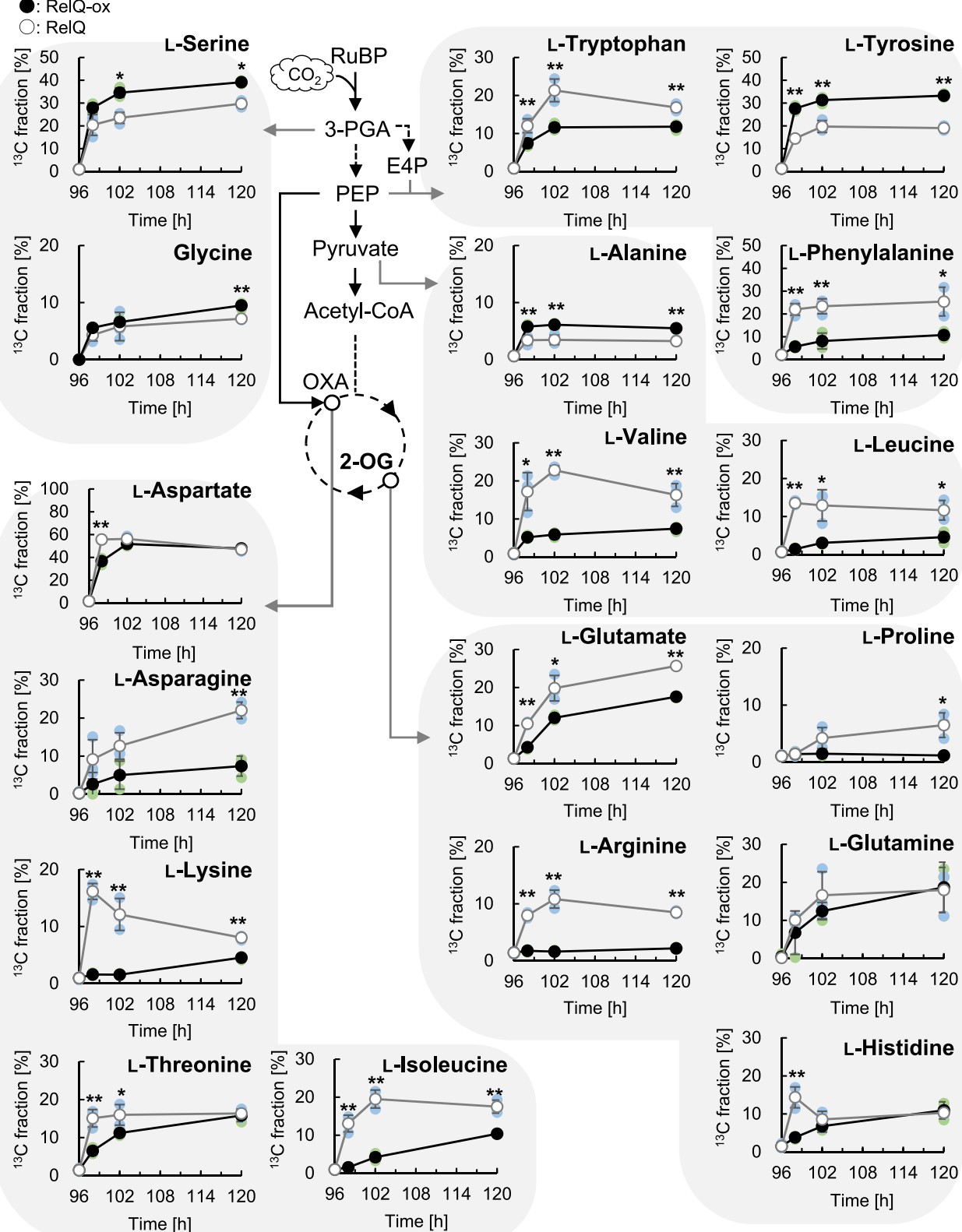

**Fig. 5 $^{13}$C-labeling pattern of amino acids in RelQ cells.** Time-course changes in $^{13}$C fractions of 18 amino acids (all except L-cysteine and L-methionine) in RelQ cells, as described in Fig. 3. Statistical significance between RelQ-ox and RelQ cells was determined using Student's $t$ test ($^{**}p < 0.01$, $^{*}p < 0.05$).

In the present study, ppGpp accumulation led to the down-regulation of genes (*ccmL*, *ccmM*, and *ccmK*) encoding carboxysome shell proteins and *rbcL/rbcS* under high $CO_2$ conditions and continuous illumination. In contrast, previous studies reported that *rbcL/rbcS* is upregulated in the RelQ-overexpressing cells under atmospheric $CO_2$ conditions[18]. One explanation for this observation is the different coordination of the metabolic signaling balance in response to $CO_2$ levels; however, the molecular

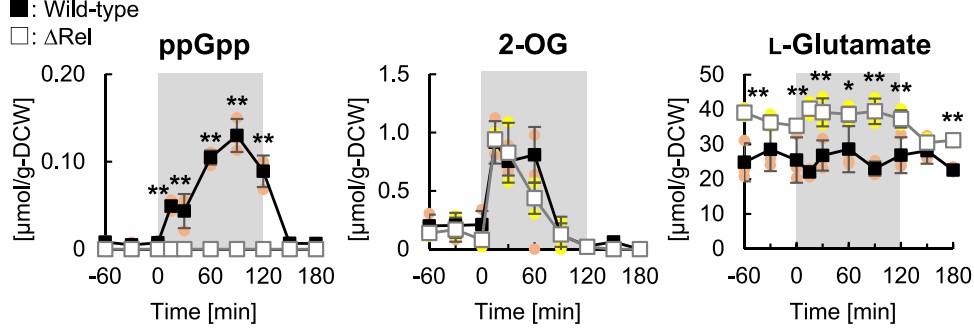

**Fig. 6 Changes in the levels of 2-OG, L-glutamate, and ppGpp in ΔRel cells during light-to-dark transition.** *S. elongatus* PCC 7942 wild type (■) and ΔRel (□) cells were cultured for 72 h with 100 μmol photons m⁻² s⁻¹ in the presence of 17 mM sodium nitrate and 1% (v/v) $CO_2$. Next, the cells were transferred to the dark and cultured for 120 min, wherein 0 min was considered the start point to the transition to dark; finally, the cells were again transferred to light conditions. The cells were harvested at 0, 15, 30, 60, 90, and 120 min and every 30 min before and after dark transitions. 2-OG, L-glutamate, and ppGpp levels were measured using CE-TOFMS analysis. All data are presented as mean ± SD (n = 3 independent biological experiments). Statistical significance between each sampling point of wild-type and ΔRel cells was determined using Student's *t* test (**$p < 0.01$, *$p < 0.05$).

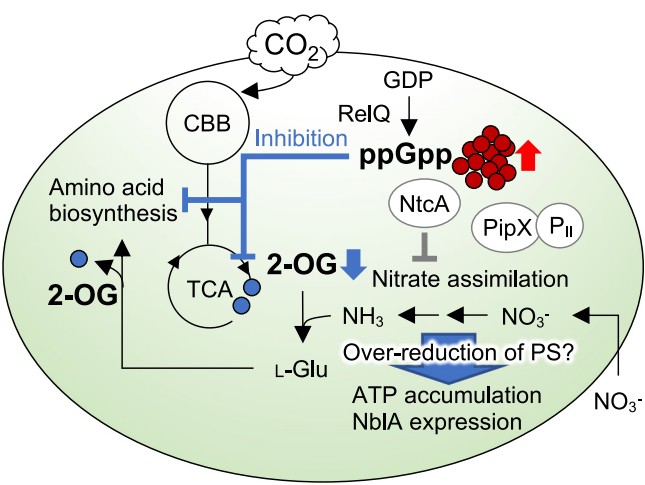

**Fig. 7 Proposed model of ppGpp signaling under photosynthetic condition.** Cellular response of *S. elongatus* PCC 7942 overexpressing RelQ. Black arrow and gray line indicate metabolic reactions and transcriptional regulation of genes for nitrogen-assimilatory enzymes, respectively. A response hierarchy of ppGpp accumulation (red circle) upon RelQ overexpression initiates lowering of 2-OG level (blue circle) through repression of 2-OG synthesis in the TCA cycle and via amino acid biosynthesis (Blue line). The PipX forms a ternary complex with 2-OG-bound NtcA required for transcription of genes involved in carbon/nitrogen regulation; PipX interacts with a P$_{II}$-signaling protein in the absence of 2-OG, resulting in an inactive-form of NtcA. Abbreviation: CBB Calvin-Benson-Bassham cycle, TCA tricarboxylic acid cycle, PS photosystem.

mechanism underlying the transcriptional regulation of ppGpp in cyanobacteria have not been elucidated. The nitrogen-assimilation pathway and $CO_2$-fixing regulated by ppGpp depend on growth conditions, such as light/dark and atmospheric $CO_2$ conditions[18]. In *E. coli*, the starvation condition-dependent role of ppGpp as a kinetic coordinator of gene expression has been revealed, highlighting its key role in global gene expression[48]. (p)ppGpp signaling may function as a molecular rheostat to couple the energy state to nutrient assimilation reactions across environmental fluctuations. Future studies should investigate the molecular mechanisms involved in metabolism and cell division that are controlled by ppGpp as natural responses to environmental changes.

## Methods

**Strains and culture conditions.** *Synechococcus elongatus* PCC 7942 and the recombinant strains were inoculated in modified medium[49], which was based on BG-11 medium[50], containing 17 mM sodium nitrate or ammonium chloride, in the presence or absence of 40 μg/mL spectinomycin and 20 μg/mL kanamycin (as required). The cells, whose initial $OD_{750}$ was set to 0.1, were cultured at 30 °C under continuous irradiation with white fluorescent light at 105–115 μE/m²/s and was supplemented with 1% (v/v) $CO_2$. The DCW in the culture was calculated based on the $OD_{750}$ value using a calibration curve. Analytical-grade chemicals were used in the study.

**Construction of recombinant strains.** The *E. coli* DH5α strain (Takara Bio, Tokyo, Japan) was used as the host for gene cloning and plasmid amplification. *relQ* gene (Uniprot ID: O31611) was amplified from *B. subtilis* strain 168 genomic DNA by PCR using the primer pair RelQ-Fw and RelQ-Rv. The amplified fragment was cloned into the PCR-amplified gene fragment containing the NSI region using the primer pair pNSHA-Fw and pNSHA-Rv from the plasmid pNSHA[51], according to the manufacturer's instructions; we used the In-Fusion HD$^R$ cloning kit (Takara Bio) (NSI[52]: between 2,578,621 bp and 2,578,662 bp, which was at the position of *Synpcc7942_2498* on the chromosome). To construct a gene cassette for *rel* (*Synpcc7942_1377*), the upstream (between 1,416,852 and 1,417,539 bp) and downstream (between 1,419,880 and 1,420,579 bp) regions were separately amplified from *S. elongatus* PCC 7942 genomic DNA by PCR using the primer pairs RELA-US-F and RELA-US-R_KM, and RELA-DS-F_KM and RELA-DS-R, respectively. A kanamycin resistance gene cassette was PCR-amplified from pUC18kan1 (NCBI accession number: LT727432) using the primer pair KM-F_RELAUS and KM-R_RELADS. The three PCR-amplified gene fragments were joined by PCR using primer sets RELA-US-F and RELA-DS-R. The principles underlying specific gene integration or disruption in *S. elongatus* PCC 7942 have been previously described[51]. The plasmid pNSI-RelQ and the PCR fragment without *rel* gene were separately used for gene integration into the host strain *S. elongatus* PCC 7942 to yield strains RelQ and ΔRel, respectively. Integration of each gene cassette was confirmed by PCR. All primer sequences are listed in Supplementary Table 1.

**Metabolome analysis.** A 5 mg-DCW portion of the *Synechococcus* cells cultured under photosynthetic or dark conditions was collected at each sampling time by filtration, using 1-μm pore size

polytetrafluoroethylene (PTFE) disks (Merck Millipore, Billerica, MA). Samples for intracellular metabolite analysis were prepared as follows, based on previously decribed[32]. Briefly, the collected samples were immediately washed with 20 mM $(NH_4)_2CO_3$ pre-cooled to 4 °C and suspended in 2 mL of pre-cooled (−30 °C) methanol containing 37.5 μM L-methionine sulfone and 37.5 μM piperazine-1,4-bis(2-ethanesulfonic acid) (PIPES) as internal standards for mass analysis. The cell suspension (0.5 mL) was mixed with 0.2 mL of pre-cooled (4 °C) water and 0.5 mL of chloroform at 4 °C and suspended by vortexing for 30 s. The aqueous and organic layers were separated by centrifugation at 14,000 × $g$ for 5 min at 4 °C. The aqueous layer (500 μL) was filtered through a 3 kDa cut-off cellulose membrane (Millipore) to remove solubilized proteins. Water was evaporated from the aqueous layer extracts under vacuum using a FreeZone 2.5 Plus freeze dryer system (Labconco, Kansas City, MO), and the dried metabolites were dissolved in 20 μL of ultrapure water. To prepare the extracellular metabolites, *Synechococcus* cells were collected at each sampling time and centrifuged at 14,000× $g$ for 5 min. The supernatant (500 μL) was mixed with 500 μL chloroform pre-cooled at 4 °C by vortexing. After centrifugation at 14,000× $g$ for 5 min at 4 °C, 400 μL of the aqueous-layer was collected and filtered using a 3 kDa cut-off cellulose membrane. Next, L-methionine sulfone and PIPES were added to the filtered aqueous solution as internal standards at final concentrations of 400 μM. Intracellular and extracellular metabolites were extracted and analyzed using CE-TOFMS (Agilent G7100; MS, Agilent G6224AA LC/MSD TOF; Agilent Technologies, Palo Alto, CA, USA) according to a previously reported method[32]. Unless otherwise noted, the intracellular pool size and extracellular concentration of each metabolite are presented as mean ± SD ($n = 3$ independent biological experiments). Statistical significance was determined using Student's $t$ test (*$p < 0.05$).

## $^{13}$C-metabolic turnover analysis

The cells were grown for 72 h as described above, followed by the addition of 1 mM IPTG. After 96 h of total cultivation, $^{13}$C-labeling was initiated by the addition of 25 mM (final concentration) sodium bicarbonate-$^{13}$C to the photoautotrophic culture medium. The culture samples were collected at the indicated time-points (0, 2, 6, and 24 h) following the addition of sodium bicarbonate-$^{13}$C. Intracellular metabolites were extracted as described above and analyzed using CE-TOFMS. The mass spectral peaks of the intermediates were identified by searching for mass shifts between $^{12}$C- and $^{13}$C-mass spectra. The $^{13}$C fractions and ratios of $^{13}$C to total carbon were calculated from the relative isotopomer abundances of metabolites incorporating $^{13}$C atoms[35]. Unless otherwise noted, the $^{13}$C fractions of each metabolite are presented as mean ± SD ($n = 3$ independent biological experiments).

## Transcriptome analysis (RNA-seq)

RelQ cells were cultured under the conditions described above, and 1 mM IPTG was added to a fraction of cells after 72 h. The cells were harvested at 78 and 96 h of total cultivation, and total RNA was isolated using the TRIzol reagent (Thermo Fisher Scientific K.K., Tokyo, Japan). cDNA libraries were constructed using the NEBNext mRNA Library Prep Master Mix Set for Illumina (NEB, Tokyo, Japan), and rRNA was removed using the Ribo-Zero rRNA Removal Kit (Illumina, Tokyo, Japan). The removal of rRNA was confirmed using an Agilent 2100 Bioanalyzer (Agilent Technologies). The cDNAs were analyzed using a MiSeq system (Illumina) with 150 cycles of paired-end sequencing (150 bases × 2). For each sample, two cDNA libraries were constructed from the RNA isolated from two biologically independent experiments, and sequencing analyses were performed twice for each cDNA library. After the

sequencing reactions were completed, the Illumina analysis pipeline (CASAVA 1.8.0) was used to process the raw sequencing data. The RNA-seq reads were trimmed using the CLC Genomics Workbench version 11.0 (Qiagen, Hilden, Germany) with the following parameters: Phred quality score >30; removing the terminal 15 nucleotides from the 5'-end and two nucleotides from the 3'-end; and discarding short reads <30 nucleotides in length. Trimmed reads were mapped to the *S. elongatus* PCC 7942 genome (NCBI accession number: NC_007604.1) using the CLC Genomics Workbench version 11.0, with the following parameters: mismatch cost, 2; indel cost, 3; length fraction, 0.7; similarity fraction, 0.9; and maximum number of hits per read, 1. Read 1 and read 2 of paired-end reads were individually mapped to the reference gene list, and the average of the two datasets was used. The expression levels of each transcript were calculated using the transcripts per million method. The TPM ratio[53] was used to identify changes in expression between cells.

## Cloning of *S. elongatus* aconitase gene and protein purification

*Synpcc7942_0903* was amplified from the genomic DNA of *S. elongatus* PCC 7942 using the primer pairs Synpcc7942_0903-Fw and Synpcc7942_0903-Rv. The amplified fragment was cloned at the NdeI/HindIII sites of the vector pET28a (Novagen, Merck Millipore, Germany), encoding an N-terminal His$_6$-tag; the obtained plasmid was designated pET-Synpcc0903. *Synpcc7942_0903* was expressed in *E. coli* BL21-CodonPlus (DE3)-RIL cells (Agilent Technologies, Santa Clara, CA), which were cultured in Luria Bertani medium containing 100 μg mL$^{-1}$ ampicillin at 37 °C for 3 h. We added 1 mM IPTG to the cells when their optical density at 600 nm reached 0.5, and the cells were cultured at 37 °C for 4 h; next, they were harvested by centrifugation at 10,000× $g$ for 30 min. The cell pellets were resuspended in buffer A (20 mM Tris-Cl, pH 7.5, and 150 mM NaCl) and disrupted by sonication in an anaerobic chamber with $O_2 < 1$ ppm (COY Laboratory Products, MI). The cell extract was centrifuged at 23,000 × $g$ for 60 min. The resultant supernatant was applied to a 5-mL HiTrap Ni-Chelating column (GE Healthcare, Little Chalfont, UK) and stepwise eluted using a gradient of imidazole (0–500 mM) in buffer A. The collected fraction was applied to a PD-10 desalting column (GE Healthcare, Little Chalfont, UK) equilibrated with Buffer A.

## Enzyme assay

RelQ cells were cultured for 72 h, then 1 mM IPTG was added, and the cells were cultured for another 24 h. The cells were suspended in 50 mM potassium phosphate buffer, disrupted by sonication, and centrifuged in an anaerobic chamber with $O_2 < 1$ ppm; the resultant supernatant was collected as the crude extract. Citrate synthase activity was aerobically measured by monitoring the amount of thiol group released from CoA-SH in a reaction mixture containing 0.1 M phosphate buffer (pH 7.6), 1 mM acetyl-CoA, 10 mM oxaloacetate, and the crude extract, using 1 mM Ellman reagent (5,5'-dithiobis); thiol group levels were determined by measuring the absorbance of the mixture at 412 nm. Aconitase assays were performed under anaerobic conditions because aconitase contains an oxygen-labile iron-sulfur cluster that is essential for its activity[54]. The aconitase activity of the crude extract was measured by monitoring the formation of NADH in a reaction mixture containing 100 mM potassium phosphate buffer (pH 8.0), 1 mM citrate, 0.1 mM NAD$^+$, 20 U isocitrate dehydrogenase, and the crude extract; NADH levels were determined by measuring the absorbance of the mixture at 340 nm. The isocitrate dehydrogenase activity of the crude extract was measured by monitoring the absorbance of a reaction mixture containing 100 mM potassium phosphate buffer (pH 8.0), 0.1 mM NAD$^+$, 1 mM isocitrate, and crude extract at 340 nm.

Protein concentrations were determined using the Bradford dye-binding assay[55]. All enzyme assays were performed at 30 °C in an V730-Bio spectrophotometer (JASCO Co., Tokyo, Japan).

The recombinant Synpcc7942_0903 protein (0.71 μM) was incubated with various concentrations of ppGpp (0.25, 0.5, 1.0, 2.5, 5.0, 7.5, and 10.0 μM) for 10 min at 30 °C. The aconitase activity of the recombinant protein incubated with different concentrations of ppGpp was measured by monitoring the formation of *cis*-aconitate in a reaction mixture containing 100 mM potassium phosphate buffer (pH 8.0) and 1 mM citrate; *cis*-aconitate levels were determined by measuring the absorbance of the mixture at 240 nm.

**qRT-PCR**. RelQ cells were cultured and harvested as described above. Total RNA was extracted from *S. elongatus* PCC 7942 frozen cells using an RNeasy Mini Kit (Qiagen) and ReverTra Ace qPCR RT Master Mix with gDNA Remover (TOYOBO, Osaka, Japan). qPCR was performed using THUNDERBIRD SYBR qPCR Mix (TOYOBO) and Mx3000P qPCR Systems (Agilent Technologies) according to the manufacturer's instructions. The primers used to amplify the selected genes by qRT-PCR are listed in Supplementary Table 1. The mRNA levels shown are the average of two independent measurements and were normalized to those of the *rnpB*-encoding RNA subunit of RNase P or the gene encoding *rimM* encoding ribosome maturation factor (reference genes), whose transcript levels did not change with ppGpp accumulation.

**Electron microscopy**. RelQ cells were cultured for 72 h as described above, followed by the addition of 1 mM IPTG; IPTG was not added to a fraction of the cells. After 96 h of IPTG addition, the cells were harvested by centrifugation and subjected to chemical fixation. The ultrastructure of RelQ cells was observed using transmission electron microscopy (Tokai Electron Microscopy, Aichi, Japan). Briefly, the cells were fixed, dehydrated with ethanol, infiltrated with propylene oxide, and polymerized in a resin. Ultrathin sections were cut, stained, and observed using a transmission electron microscope (JEM-1400Plus; JOEL Ltd., Tokyo, Japan) at an acceleration voltage of 80 kV. Digital images (2048 × 2048 pixels) were captured using a CCD camera (VELETA; Olympus Soft Imaging Solutions, Münster, Germany).

**Other analytical methods**. Glycogen concentration was determined by measuring the amount of glucose released from glycogen by enzymatic hydrolysis[32]. The amount of glucose was evaluated using an HPLC system equipped with an Aminex HPX-87 column and a refractive index detector. To determine the residual nitrate concentration in the culture medium, cyanobacterial cultures were centrifuged at 5000× *g* for 1 min, and the absorbance of the supernatant, which was diluted 20-folds using distilled water, was measured at 220 nm, where NaNO₃ exhibited maximum absorbance[56], using a UV mini-1240 UV-Vis spectrophotometer (Shimadzu, Kyoto, Japan). The residual nitrate content was determined using a calibration curve. The residual ammonium concentration in the culture medium was determined using LabAssay™ Ammonia (Wako Pure Chemical Industries, Osaka, Japan) according to the manufacturer's protocol.

**Statistics and reproducibility**. In the present study, all experiments, except for RNA-Seq and qPCR analyses, were basically conducted using three independent biological replicates. The average value and the standard deviation of the individual data points were calculated and visualized using Microsoft Excel software. The statistical significance was calculated with Student's *t*-test and the Dunnett test by using R program.

**Reporting summary**. Further information on research design is available in the Nature Portfolio Reporting Summary linked to this article.

## Data availability

All data supporting the findings of this study are available within the paper and its Supplementary Information/Data. Raw data of the figures can be found in Supplementary Data 1. DNA sequence data can be found in Supplementary Data 2. The raw sequence data of RNA-seq analysis can be found in the Genome Sequence Archive database (https://ngdc.cncb.ac.cn/gsa); the assigned accession is CRA013529 (https://bigd.big.ac.cn/gsa/browse/CRA013529). Unedited images for the gel in Fig. 4b and Supplemental Fig. 1b can be found in Supplemental Fig. 6. All other data are available from the corresponding author on reasonable request.

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

## Acknowledgements
The authors thank Ms. Hiroko Koizumi for their technical assistance. The RNA sequencing was performed by Research Institute of Green Science and Technology, Shizuoka University. This work was supported by Japan Science and Technology Agency (JST)-Mirai Program Grant Number JPMJMI19E4, the Ministry of Education, Culture, Sports, Science, and Technology (MEXT), Japan.

## Author contributions
R.H. and R.O. contributed equally. R.O., H.A., and T.H. conceived and designed the research. R.H., R.O., M.M, and Y.K. performed most experiments and M.M. conducted CE-TOFMS analysis. S.I. and K.T. contributed to the interpretation of the results. R.H., R.O., Y.K., H.A., and T.H. analyzed data. A.K. and T.H. supervised the research. R.H., Y.K., R.O., H.A., and T.H. wrote the manuscript. All authors approved the final version of the manuscript.

## Competing interests
The authors declare no competing interests.
