## [Peer Review File · Communications Biology]

Reviewers' comments:

Reviewer #1 (Remarks to the Author):

ppGpp is an important signaling molecule through its function in the regulation of transcription, translation and DNA replications processes in bacteria. As a global regulator, its functions in cyanobacteria are relatively poorly understood. Nevertheless, several publications demonstrate that it is likely essential or very critical for cell survival. In the current manuscript, the authors used an over expression strain of the unicellular cyanobacterium *Synechococcus* PCC 7942, to assess the impact of ppGpp accumulation on cell physiology. The authors found that cells displayed a N starvation-like response, among others. Many of the function may be explained by the higher level of 2-OG, a carbon skeleton serving as a N/C balance signal. Others are poorly explained, which may correspond to the indirect unknown effect of ppGpp.

1) Major comments. One obvious weakness is the lack a mutant with no ppGpp, or lower ppGpp. Note that such high level of ppGpp seen here is an artificial situation created by the over expression of ppGpp synthétase, something that may never occur in the real situation. Even a partially segregated mutant would have been useful. In the current situation, it would have been important that the authors compare their data with those published data using a ppGpp-deficient strain in the same organism, so that a confrontation of data from ppGpp overproduction with those from lacking ppGpp published by others. Otherwise, it would be hard to deduce which are the responses that are physiologically meaningful.

2) How to explain the effect on 2-OG?

2) For other minor problems, I list below my comments.

Lines 61. It is too simplistic to say that darkness is analogous to nutrient starvation. Cyano adapts to light-dark cycle even abrupt situation, and may indirectly impact on nutrient assimilation. There are also references showing the function of ppGpp in heterocyst formation, which is related to N/C balance (Akinyanju J, Smith RJ. *New Phytol.* 1987 105(1):117-122. Zhang et al., *J Bacteriol.* 2013 Oct;195(19):4536-44). Which needs to be cited and discussed.

Paragraph starting with line 72. In the whole section, key literatures about 2-OG signaling are lacking: on the identification of 2-OG as a signal in vivo (Laurent et al., *PNAS*, 2005, 102:9907-9912); Structural evidence (Zhao et al., *PNAS* 2010, 107(28):12487-92; Ll acer JL et al., *PNAS* 2010, 107(35):15397-402).

Line 109: what does it mean "reached about 37%..."?

Lines 143-144 and discussion, if some of the CCM genes are negatively affected, why then do cells accumulate glycogen ?

Figs 2 and 3. It is known that 2-phosphoglycolate is a N/C balance signal (Jiang et al., *PNAS*, 2018, 115:403-408; Zhang et al., *Trends in Plant Sci.*, 2018, 23:1116-1130). Have the authors checked the level of this metabolite ?

Line 224-225. On the effect on 2-OG synthesis. Some of the TCA cycle genes have a NtcA binding box. Have the authors checked that ? It could be a result of feedback regulation.

Lines 264-265. I don't see result of a stress.

Lines 305-306. It is unlikely that ppGpp pathway directly senses N/C balance. More likely a consequence of global regulation.

Reviewer #2 (Remarks to the Author):

The MS by Hidese et al describe transcriptome (with the emphasis in the NtcA regulon) and metabolomic results obtained in the cyanobacterium *Synechococcus elongatus* PCC7942 previously inundated of ppGpp. For this they cloned and subsequently introduce into the cyanobacterium the gene encoding the small (p)ppGpp synthetase YjbM/SAS1/RelQ under the control of the IPTG-inducible promoter Ptrc. This is exactly the same strategy and model system used to investigate ppGpp roles in cyanobacteria in the very insightful work by Hood et al 2016, where they performed comprehensive and very careful transcriptomic (but not metabolomic), analysis, explicitly justifying experimental settings and being far more cautious with conditions (OD of starting cultures, IPTG concentrations, times of treatment, etc.) to ensure the physiological relevance of the data.

I think that it is obliged that Hidese et al explain what is now new or important to justify revisitation of an apparently very well performed transcriptomic analysis that was published years ago.

There is also a very recent paper on the phenotypic effects of overexpression of YjbM/SAS1/RelQ in *Synechococcus elongatus* PCC7942, that should be taken into account. Llop et al 2023 use the construct from Hood et al 2016 and refer to the protein as RelQ. Since YjbM/SAS1/RelQ it is not a hypothetical protein, RelQ is a more correct denomination than YjbM, and thus using RelQ would simplify things to potential readers.

A main conclusion of the present work is that under a very prolonged and non-physiological excess of ppGpp, the levels of 2-oxoglutarate are very low, and that is "probably because ppGpp caused inhibition of aconitase activity". This is an interesting result, but I am not sure that the in vitro experiment to show direct inhibition of the enzyme is that sound.

I find the MS difficult to read, lacking precision (even in the abstract) and organization. I do not understand at all why there are two sections of text with results and discussion in Supplementary.

I also find very confusing that the only *Synechococcus elongatus* PCC7942 strain under study is called "YjbM cells" and then in the experiments/figures "YjbM- cells" or "YjbM+ cells" according to the addition or not of IPTG. I think it is not correct because the Ptrc promoter is very leaky (please check literature) and the "YjbM- cells" do have increased ppGpp levels, as explicitly shown in the previous papers with RelQ constructs. So, I think it would be more correct to say RelQ and RelQI (or RelQox) cells to indicate that the difference refers to induction (or overexpression) triggered by IPTG and not to the presence or absence of a gene/protein, as it is used with endogenous genes to name phenotypes in classical genetic analysis.

The writing is not clear, with many confusing sentences and incorrect expressions, like in line 68 "the (p)ppGpp deletion ..." or in line 84 "heterogeneous expression"

Reviewer #3 (Remarks to the Author):

In principal, this paper reports the interesting observation that elevated ppGpp levels lead to a reduced de novo synthesis of 2-oxoglutarate in the cyanobacterium *Synechococcus elongatus*. Since 2-OG is an important metabolite status reporter that affects sensing of the homeostatic system, this is an important new insight. To substantiate their data, the authors have searched for the molecular basis of reduced 2-OG synthesis and they identified Aconitase as target of ppGpp action. In vitro the authors could demonstrate that ppGpp tunes down the activity of this enzyme, albeit that the degree of inhibition is only moderate. It would be helpful for the understanding of the data if the authors could calculate or model, if this moderate decrease in aconitase activity is sufficient to explain the severe reduction of de novo 2-OG synthesis that they reported by C13 labelling experiments. If the

degree of inhibition is not sufficient to explain the reduced flux, additional regulatory points of ppGpp need to be taken into consideration. I would not be surprised if this would be the case, since the metabolism is quite complex and coherent regulation can occur at different places.

Besides this, there is an odd misconception of what is a nutrient and what is energy source. This misconception impairs the entire paper. Already in the introduction, the authors equate "light limitation" with "nutrient limitation" (line 61-61). Light is not a nutrient! Light is the energy source for phototrophs. A fundamental concept in microbiology classifies the organisms according to their energy and nutrient demand. According to the energy source of the organisms, we distinguish between "Phototrophs", like Cyanobacteria (using light energy as primary energy source) and "Chemotrophs" (using a chemical reaction such as respiration as primary energy source). Only within the chemotrophs, we further distinguish between Chemolithotrophs (using inorganic substrates) and Chemoorganotrophs (using organic substrates)

According to the nutrient demand, we distinguish between "Autotrophs" (using inorganic carbon as primary carbon nutrient) and "Heterotrophs" (using organic nutrients). A nutrient is per definition a substance, a molecule. So, the statement that light limitation is analogous to "nutrient limitation" is completely odd. It neglects the metabolic concept of cyanobacteria, where energy metabolism by photosynthetic electron transport is separated from the availability of "nutrients". Availability of nutrients for energy production is only true in the case in heterotrophic bacteria. Only there, energy supply depends on the availability of nutrients.

As for the case of stringent response, it appears that in cyanobacteria this process responds primarily to the energy state of the cells, modulated by the availability of photosynthetically usable photons. The assimilatory reactions such as CO₂ fixation and nitrogen assimilation depend on the reducing power and energy that is generated by photosynthesis. The limitation of C and N nutrients may even lead to an excess of energy, since the assimilatory reactions that dissipate most of the photosynthetically acquired energy are now limiting. Therefore, it is extremely important not to mix up "darkness" with "nutrient deprivation" – these are two fundamentally different situations.

Minor:

I. 75: you state that "light deprivation" (do you mean shift to darkness?) results in 2-OG accumulation. The references that are given here do not support this sentence. Moreover, I'm not aware that it has ever been shown that 2-OG levels increase when cells are shifted to darkness. According to your own discovery, the contrary should be the case: as darkness leads to increased ppGpp levels, a reduction in 2-OG levels is expected.

L. 305 ff: The following sentence confuses cause and effect: "(p)ppGpp signaling may function as a molecular rheostat by sensing the C/N balance and energy charge ratio across a spectrum of environmental conditions ranging from nutrient rich to nutrient poor" : Your work reveals how ppGpp, by responding to dark-shift, couples the energy state to assimilatory reactions (through modulating the status reporter 2-OG) and thereby regulates the C/N balance. However, there is no evidence that ppGpp SENSES the C/N balance. C/N sensing is achieved by sensing 2-OG levels.

Methods: the description of metabolite analysis is incomplete. In particular, determination of extracellular metabolites needs careful description (like extraction from medium ...)

A very recent paper describes phenotypic changes in *Synechococcus*, imposed by overexpression of the ppGpp synthase RelQ (the equivalent of YjbM) (Frontiers in Microbiology, March 2023). Although the focus of this study was different (as RelQ served as a control) the similarities should be briefly mentioned.

Besides of these above-mentioned drawbacks, the current paper makes an important contribution by identifying a reaction where ppGpp affects metabolism. I think, the paper can be revised without need for new experiments.

Referee expertise:

Referee #1: Cyanobacteriology

Referee #2: Molecular microbiology

Referee #3: Nitrogen regulation in bacteria

ppGpp is an important signaling molecule through its function in the regulation of transcription, translation and DNA replications processes in bacteria. As a global regulator, its functions in cyanobacteria are relatively poorly understood. Nevertheless, several publications demonstrate that it is likely essential or very critical for cell survival. In the current manuscript, the authors used an over expression strain of the unicellular cyanobacterium *Synechococcus* PCC 7942, to assess the impact of ppGpp accumulation on cell physiology. The authors found that cells displayed a N starvation-like response, among others. Many of the function may be explained by the higher level of 2-OG, a carbon skeleton serving as a N/C balance signal. Others are poorly explained, which may correspond to the indirect unknown effect of ppGpp.

1) Major comments. One obvious weakness is the lack a mutant with no ppGpp, or lower ppGpp. Note that such high level of ppGpp seen here is an artificial situation created by the over expression of ppGpp synthétase, something that may never occur in the real situation. Even a partially segregated mutant would have been useful. In the current situation, it would have been important that the authors compare their data with those published data using a ppGpp-deficient strain in the same organism, so that a confrontation of data from ppGpp overproduction with those from lacking ppGpp published by others. Otherwise, it would be hard to deduce which are the responses that are physiologically meaningful.

Thank you for your valuable comments. We agree with your comment that the authors compare our data with those published data including reports from Hood et al (Proc Natl Acad Sci U S A. 113(33), E4867-76 (2016)), Puszynska et al. (Cell Rep. 21(11):3155-3165 (2017)), and Liop et al. (Front Microbiol. 14, 1141775 (2023)). To confirm the regulation of 2-OG level as the natural response, we knocked-out the *rel* gene from *S. elongatus* PCC7942 wild-type, resulting in the Δ Rel strain. No ppGpp production was observed in the Δ Rel strain. Accumulation of ppGpp peaked at 90 min after dark transition in wild-type strain, whose ppGpp level was comparable to that of RelQ-ox strain. Interestingly, the Δ Rel strain accumulated L-glutamate, irrespective of light-to-dark transition, suggesting that ppGpp regulates L-glutamate level through repression of de novo synthesis of 2-OG.

The newly obtained data were added to the revised manuscript in lines 295-313, page 16 and figure 6.

2) How to explain the effect on 2-OG?

We showed that ppGpp negatively regulates the activity of aconitase that catalyzes conversion of cis-aconitate to 2-oxoglutarate. However, the inhibition effect of ppGpp on aconitase activity was not significant to explain why ¹³C-labeling of 2-OG is low in RelQ-ox cells. The metabolic checkpoint by ppGpp should be not only in de novo -OG synthesis but also in other primary metabolites. In fact, the ¹³C-labeling of many amino acids including L-leucine, L-valine, L-isoleucine, L-histidine, L-phenylalanine, and L-aspartate, for which their biosyntheses depend on glutamate-dependent aminotransferases, were decreased in the RelQ-ox cells (in lines 265-282, page 13). We discussed the ppGpp effect on 2-OG biosynthesis in lines 343-349, pages 17-18.

2) For other minor problems, I list below my comments.

Lines 61. It is too simplistic to say that darkness is analogous to nutrient starvation. Cyano adapts to light-dark cycle even abrupt situation, and may indirectly impact on nutrient assimilation. There are also references showing the function of ppGpp in heterocyst formation, which is related to N/C balance (Akinyanju J, Smith RJ. *New Phytol.* 1987 105(1):117-122. Zhang et al., *J Bacteriol.* 2013 Oct;195(19):4536-44). Which needs to be cited and discussed.

Thank you for your valuable comment. As reviewer 2 comment, light is energy source, but not nutrient source. Darkness imposes cyanobacterium energy deprivation, resulting in the impairment of nutrient metabolisms such as CO₂ fixation and nitrogen assimilation. We have rewritten the manuscript in lines 64-66, page 3. We also added the references as you suggested and discussed the role of ppGpp in the heterocyst formation in lines 332-334, in page 17.

Paragraph starting with line 72. In the whole section, key literatures about 2-OG signaling are lacking: on the identification of 2-OG as a signal in vivo (Laurent et al., *PNAS*, 2005, 102:9907-9912); Structural evidence (Zhao et al., *PNAS* 2010, 107(28):12487-92; Ll acer JL et al., *PNAS* 2010, 107(35):15397-402).

Thank you for your suggestion. We have added the references in the suitable section (line 83, page 4) in the revised manuscripts.

Line 109: what does it mean "reached about 37%..."?

Thank you for your suggestion. We have rewritten the sentence to adequately address our results in line 114, page 5.

Lines 143-144 and discussion, if some of the CCM genes are negatively affected, why then do cells

accumulate glycogen ?

Our ¹³C-labeling data showed that the rates of ¹³C-labeling of CBB intermediates, including F6P and 3PGA, of RelQ-ox cells were comparable to that of RelQ cells within 24 h. This indicates that CO₂-fixation continues in RelQ-ox cells under high CO₂ concentration, even though the transcript level of CCM was down-regulated. We think the CCM transcript level is sufficient for carbon fixation in RelQ-ox cells. Our metabolome analysis revealed that the pool size of ATP increased in time-dependent manner. The fixed carbon would be glycogen to consume the ATP.

Figs 2 and 3. It is known that 2-phosphoglycolate is a N/C balance signal (Jiang et al., PNAS, 2018, 115:403-408; Zhang et al., Trends in Plant Sci., 2018, 23:1116-1130). Have the authors checked the level of this metabolite ?

The 2-phosphoglycolate was below the detection limit in our CE-MS system, therefore, we could not quantify 2-phosphoglycolate.

Line 224-225. On the effect on 2-OG synthesis. Some of the TCA cycle genes have a NtcA binding box. Have the authors checked that ? It could be a result of feedback regulation.

Thank you for giving us kind advice. As you mentioned, a TCA cycle enzyme isocitrate dehydrogenase gene is a known NtcA target (Giner-Lamia J., et al. Nucleic Acids Res. 2017 Nov 16;45(20):11800-11820). Our RNA-seq analysis showed that the TCA cycle gene *icd* (Synpcc7942_1719) under the regulation of NtcA were not significantly changed in the RelQ-ox cells (0.152: 6 h, -0.352: 24 h). Our in vitro assay revealed that isocitrate dehydrogenase activity was comparable between Rel and Rel-ox cells, indicating that IDH is not the target for ppGpp inhibition.

Lines 264-265. I don't see result of a stress.

ppGpp accumulation is widely known to be occurred in stress conditions, such as nitrogen starvation and light deprivation. In the present study, we focused on the effect of ppGpp on metabolism by RelQ overexpression. As the reviewer pointed out, to address the contents of our results adequately, the sentence "that enabled us to visualize the cellular response by transient ppGpp accumulation under stress conditions" was omitted in the present manuscript in lines 326-327, page 17.

Lines 305-306. It is unlikely that ppGpp pathway directly senses N/C balance. More likely a consequence of global regulation.

Thank you for your kind suggestion. We have rewritten the manuscripts to adequately address our discussion in lines 379-380, page 19.

Reviewer #2 (Remarks to the Author):

The MS by Hidese et al describe transcriptome (with the emphasis in the NtcA regulon) and metabolomic results obtained in the cyanobacterium *Synechococcus elongatus* PCC7942 previously inundated of ppGpp. For this they cloned and subsequently introduce into the cyanobacterium the gene encoding the small (p)ppGpp synthetase YjbM/SAS1/RelQ under the control of the IPTG-inducible promoter Ptrc. This is exactly the same strategy and model system used to investigate ppGpp roles in cyanobacteria in the very insightful work by Hood et al 2016, where they performed comprehensive and very careful transcriptomic (but not metabolomic), analysis, explicitly justifying experimental settings and being far more cautious with conditions (OD of starting cultures, IPTG concentrations, times of treatment, etc.) to ensure the physiological relevance of the data.

I think that it is obliged that Hidese et al explain what is now new or important to justify revisitation of an apparently very well performed transcriptomic analysis that was published years ago. There is also a very recent paper on the phenotypic effects of overexpression of YjbM/SAS1/RelQ in *Synechococcus elongatus* PCC7942, that should be taken into account. Llop et al 2023 use the construct from Hood et al 2016 and refer to the protein as RelQ. Since YjbM/SAS1/RelQ it is not a hypothetical protein, RelQ is a more correct denomination than YjbM, and thus using RelQ would simplify things to potential readers. A main conclusion of the present work is that under a very prolonged and non-physiological excess of ppGpp, the levels of 2-oxoglutarate are very low, and that is “probably because ppGpp caused inhibition of aconitase activity”. This is an interesting result, but I am not sure that the in vitro experiment to show direct inhibition of the enzyme is that sound.

I find the MS difficult to read, lacking precision (even in the abstract) and organization. I do not understand at all why there are two sections of text with results and discussion in Supplementary. I also find very confusing that the only *Synechococcus elongatus* PCC7942 strain under study is called “YjbM cells” and then in the experiments/figures “YjbM- cells” or “YjbM+ cells” according to the addition or not of IPTG. I think it is not correct because the Ptrc promoter is very leaky (please check literature) and the “YjbM- cells” do have increased ppGpp levels, as explicitly shown in the previous papers with RelQ constructs. So, I think it would be more correct to say RelQ and RelQI (or RelQox) cells to indicate that the difference refers to induction (or overexpression) triggered by IPTG and not to the presence or absence of a gene/protein, as it is used with endogenous genes to name phenotypes in classical genetic analysis.

The writing is not clear, with many confusing sentences and incorrect expressions, like in line 68 “the (p)ppGpp deletion ...” or in line 84 “heterogeneous expression”

Thank you for valuable comments on our manuscript. As the reviewer suggested, we renamed YjbM as RelQ and more correct to say RelQ and RelQ-ox as a whole, since RelQ strain contained ppGpp in a basal level. The report from Hood et al. provided us very important data and insightful discussion in ppGpp roles in cyanobacteria. In fact, the results obtained by our RNA-seq

analysis were almost consistent with the reports from Hood et al. For example, the downregulation of the genes (nitrate reductase genes and nitrate transporter gene) for nitrate assimilation was also observed in the previous report. However, the mechanistic insight into the downregulation of the genes for nitrate assimilation has not been fully discussed in the previous manuscript. Here, we present novel finding by using dynamic metabolomic approach, in addition to our transcriptome data. The most important finding is that RelQ-overexpression decreased intracellular 2-oxoglutarate level, probably due to enzymatic inhibition by ppGpp in metabolism. The low level of 2-OG leads to deactivation of a nitrogen global transcriptional regulator NtcA, resulting in the repression of nitrogen assimilation. To make our novel finding clear, we have rewritten the sections of abstract (lines 34-37, page 2), introduction (lines 74-85, pages 3-4) and results (lines 110-112, page 5; lines 127-139, pages 5-6).

The low ^{13}C -labelling of 2-oxoglutarate by RelQ-overexpression seems not to be explained simply by inhibition of aconitase, because the inhibition rate of aconitase activity was moderate when considering difference in the ^{13}C -labeling rates of 2-OG between RelQ and RelQ-ox cells. 2-OG is mainly synthesized through TCA cycle, and the one is re-produced by biosyntheses of several amino acids including L-valine, L-isoleucine, L-leucine, L-histidine, L-phenylalanine, L-tyrosine, and L-aspartate through corresponding aminotransferase reactions as the byproduct. The present ^{13}C -labeling data showed low ^{13}C -labeling of these amino acids, except for L-tyrosine, in RelQ-ox cells.

To address the Editor and reviewer's concern, we analyzed intracellular 2-OG and L-glutamate concentration in a deletant of rel gene in the cyanobacterium. As the reviewer pointed out, the ppGpp level by RelQ overexpression is ten times higher than those in wild-type strain, which is not physiologically relevant. However, we observed that accumulation of ppGpp peaked at 90 min after dark transition in wild-type strain, whose ppGpp level was comparable to that of RelQ-ox strain. The ppGpp-deficient strain ΔRel accumulated L-glutamate, which is synthesized from 2-OG, irrespective of light-to-dark transition, when compared to that of wild-type strain. The newly obtained data (lines 295-313, page 16) supported our insight that ppGpp negatively controls intracellular 2-oxoglutarate level. We checked our manuscript after proofreading of a commercial English editing service and handled it appropriately.

Reviewer #3 (Remarks to the Author):

In principal, this paper reports the interesting observation that elevated ppGpp levels lead to a reduced de novo synthesis of 2-oxoglutarate in the cyanobacterium *Synechococcus elongatus*. Since 2-OG is an important metabolite status reporter that affects sensing of the homeostatic system, this is an important new insight. To substantiate their data, the authors have searched for the molecular basis of reduced 2-OG synthesis and they identified Aconitase as target of ppGpp action. In vitro the authors could demonstrate that ppGpp tunes down the activity of this enzyme, albeit that the degree of inhibition is only moderate. It would be helpful for the understanding of the data if the authors could

calculate or model, if this moderate decrease in aconitase activity is sufficient to explain the severe reduction of de novo 2-OG synthesis that they reported by C13 labelling experiments. If the degree of inhibition is not sufficient to explain the reduced flux, additional regulatory points of ppGpp need to be taken into consideration. I would not be surprised if this would be the case, since the metabolism is quite complex and coherent regulation can occur at different places.

Besides this, there is an odd misconception of what is a nutrient and what is energy source. This misconception impairs the entire paper. Already in the introduction, the authors equate “light limitation” with “nutrient limitation” (line 61-61). Light is not a nutrient! Light is the energy source for phototrophs. A fundamental concept in microbiology classifies the organisms according to their energy and nutrient demand. According to the energy source of the organisms, we distinguish between “Phototrophs”, like Cyanobacteria (using light energy as primary energy source) and “Chemotrophs” (using a chemical reaction such as respiration as primary energy source). Only within the chemotrophs, we further distinguish between Chemolitotrophs (using inorganic substrates) and Chemoorganotrophs (using organic substrates)

According to the nutrient demand, we distinguish between “Autotrophs” (using inorganic carbon as primary carbon nutrient) and “Heterotrophs” (using organic nutrients). A nutrient is per definition a substance, a molecule. So, the statement that light limitation is analogous to “nutrient limitation” is completely odd. It neglects the metabolic concept of cyanobacteria, where energy metabolism by photosynthetic electron transport is separated from the availability of “nutrients”. Availability of nutrients for energy production is only true in the case in heterotrophic bacteria. Only there, energy supply depends on the availability of nutrients.

As for the case of stringent response, it appears that in cyanobacteria this process responds primarily to the energy state of the cells, modulated by the availability of photosynthetically usable photons. The assimilatory reactions such as CO₂ fixation and nitrogen assimilation depend on the reducing power and energy that is generated by photosynthesis. The limitation of C and N nutrients may even lead to an excess of energy, since the assimilatory reactions that dissipate most of the photosynthetically acquired energy are now limiting. Therefore, it is extremely important not to mix up “darkness” with “nutrient deprivation” – these are two fundamentally different situations.

Thank you very much for your valuable comments. Because the aconitase activity of crude extract in the Rel-ox cells was about one-half of that of Rel cells, while, the aconitase activity (specific activity) directly inhibited by ppGpp was up to 60% of relative activity, the reduction of aconitase activity by ppGpp may be due to quantitative adjustment of aconitase in addition to direct regulation of the activity by ppGpp. If de novo synthesis of 2-OG depends solely on aconitase activity, the ¹³C-labeling rate of 2-OG must be comparable to inhibition rate of aconitase in a ppGpp-concentration-dependent manner. Whereas, ¹³C-labeling rate of 2-OG in the RelQ cells was ten-time higher than those in the RelQ-ox cells, suggesting that total ¹³C-labeling rate of 2-OG was due to the other

regulatory points in addition of aconitase, as you pointed out. The 2-OG is produced not only by de novo synthesis via TCA cycle but also by both transamination reactions for amino acid biosynthesis and glutamate dehydrogenase. The low ¹³C-labeling of amino acids would partially contribute to the low ¹³C-labeling of 2-OG in terms of turn-over of metabolites. For example, the ¹³C-labelings of the branched-chain amino acids L-isoleucine and L-leucine in RelQ-ox cells were ten-times lower than those in RelQ cells. We have rewritten the results of ¹³C-labeling analysis in detail in lines 265-282, page 13.

As for the statement that light limitation is analogous to “nutrient limitation”, we have corrected the previous statement to “Dark conditions limit the availability of external energy sources for cyanobacteria, leading to metabolic rearrangement, i.e., changes in carbon and nitrogen fixation pathways” (lines 59-62, page 3).

Minor:

l. 75: you state that “light deprivation” (do you mean shift to darkness?) results in 2-OG accumulation. The references that are given here do not support this sentence. Moreover, I’m not aware that it has ever been shown that 2-OG levels increase when cells are shifted to darkness. According to your own discovery, the contrary should be the case: as darkness leads to increased ppGpp levels, a reduction in 2-OG levels is expected.

Thank you for your comments. We apologize for presenting unsuitable reference. As you mentioned, there are no obvious results the 2-OG level increase when cells are shifted to darkness. Therefore, we have deleted the sentence on the basis of uncertain information.

L. 305 ff: The following sentence confuses cause and effect: “(p)ppGpp signaling may function as a molecular rheostat by sensing the C/N balance and energy charge ratio across a spectrum of environmental conditions ranging from nutrient rich to nutrient poor“ : Your work reveals how ppGpp, by responding to dark-shift, couples the energy state to assimilatory reactions (through modulating the status reporter 2-OG) and thereby regulates the C/N balance. However, there is no evidence that ppGpp SENSES the C/N balance. C/N sensing is achieved by sensing 2-OG levels.

Thank you for valuable comments. As you pointed out, we have rewritten the description in lines 379-380, page 19.

Methods: the description of metabolite analysis is incomplete. In particular, determination of extracellular metabolites needs careful description (like extraction form medium ...)

We have rewritten the description of metabolite analysis in detail in lines 426-441, page 21.

A very recent paper describes phenotypic changes in *Synechococcus*, imposed by overexpression of

the ppGpp synthase RelQ (the equivalent of YjbM) (Frontiers in Microbiology, March 2023). Although the focus of this study was different (as RelQ served as a control) the similarities should be briefly mentioned.

We mentioned the recent paper that demonstrates phenotypic changes in the *Synechococcus* (Introduction section, lines 69-71, page 3; Result section, lines 110-112, page 5).

Besides of these above-mentioned drawbacks, the current paper makes an important contribution by identifying a reaction where ppGpp affects metabolism. I think, the paper can be revised without need for new experiments.

Reviewers' comments:

Reviewer #1 (Remarks to the Author):

The authors addressed most of my concerns in the revised manuscript. I have no other comments to add.

Reviewer #2 (Remarks to the Author):

Thank you for considering my suggestions.

Reviewer #3 (Remarks to the Author):

It is obvious that the authors made great efforts to further strengthen their paper. Unfortunately, the new data do not clarify the contradictions in this paper. The authors also ignored the comments regarding the significance of ppGpp signaling in cyanobacteria. Fig 6 shows clearly that light to dark transition is a major inducer of ppGpp accumulation under physiological conditions. It is also obvious from this experiment, that this ppGpp induction has no effect on 2-OG accumulation: the increased of 2-OG is almost identical in wild type and relQ deficient mutant. The RelQ over expression experiment is a rather artificial setting, and it is still not clear if the observed effects are relevant form ppGpp signaling under physiological conditions (which its mainly dark transition).

Reviewers' comments:

Reviewer #3 (Remarks to the Author):

It is obvious that the authors made great efforts to further strengthen their paper. Unfortunately, the new data do not clarify the contradictions in this paper. The authors also ignored the comments regarding the significance of ppGpp signaling in cyanobacteria. Fig 6 shows clearly that light to dark transition is a major inducer of ppGpp accumulation under physiological conditions. It is also obvious from this experiment, that this ppGpp induction has no effect on 2-OG accumulation: the increased of 2-OG is almost identical in wild type and relQ deficient mutant. The RelQ over expression experiment is a rather artificial setting, and it is still not clear if the observed effects are relevant form ppGpp signaling under physiological conditions (which its mainly dark transition).

Thank you very much for giving us your very important comments on our paper. Certainly, as you point out, it is unclear whether ppGpp is induced under physiological conditions except for light-to-dark transition. However, it is possible, for example, that if 2-OG increases too much above a certain threshold, ppGpp may show a feedback effect. The results in Fig. 6 clearly show that the inhibition of novel 2-OG synthesis by RelQ overexpression is a phenomenon that does not occur under dark conditions. This may be due to the fact that carbon fixation is restricted under dark conditions, affecting their carbon partitioning to the TCA cycle. Indeed, this threshold is rarely exceeded in the dark, but ppGpp may play a role in preventing 2-OG levels from being exceeded during nitrogen deprivation or another starvation stress. Therefore, the present RelQ overexpression experiment observes the metabolic variability and flexibility to survive, due to artificial ppGpp accumulation under light conditions. On the other hand, the amount of ppGpp at basal level suppressed L-glutamate synthesis, which is synthesized from 2-OG as a precursor, under light conditions. We believe that the effect of ppGpp accumulation on the variation of metabolisms of 2-OG derivatives has important implications to elucidate yet-unknown various ppGpp signaling mechanisms. With this in mind, we have added an explanation to clarify the significance of our findings in the ppGpp signaling research in lines 333-338, page 17.